artificial intelligence/environmental science/ computer vision

city similarity, city streetscape, deep learning, street-level imagery

**Author for correspondence:**
Yu Liu
e-mail: liuyu@urban.pku.edu.cn

# Discovering place-informative scenes and objects using social media photos

Fan Zhang[1,3], Bolei Zhou[2], Carlo Ratti[3] and Yu Liu[1]

[1]Institute of Remote Sensing and Geographical Information Systems, School of Earth and Space Sciences, Peking University, Beijing 100871, People's Republic of China
[2]Department of Information Engineering, The Chinese University of Hong Kong, Hong Kong, People's Republic of China
[3]Senseable City Laboratory, Massachusetts Institute of Technology, MA 02139, USA

(iD) FZ, 0000-0002-3643-018X; YL, 0000-0002-0016-2902

Understanding the visual discrepancy and heterogeneity of different places is of great interest to architectural design, urban design and tourism planning. However, previous studies have been limited by the lack of adequate data and efficient methods to quantify the visual aspects of a place. This work proposes a data-driven framework to explore the place-informative scenes and objects by employing deep convolutional neural network to learn and measure the visual knowledge of place appearance automatically from a massive dataset of photos and imagery. Based on the proposed framework, we compare the visual similarity and visual distinctiveness of 18 cities worldwide using millions of geo-tagged photos obtained from social media. As a result, we identify the visual cues of each city that distinguish that city from others: other than landmarks, a large number of historical architecture, religious sites, unique urban scenes, along with some unusual natural landscapes have been identified as the most place-informative elements. In terms of the city-informative objects, taking vehicles as an example, we find that the taxis, police cars and ambulances are the most place-informative objects. The results of this work are inspiring for various fields—providing insights on what large-scale geo-tagged data can achieve in understanding place formalization and urban design.

## 1. Introduction

The physical setting of a place, such as a cityscape, is shaped heterogeneously by the difference in the development of culture, geography and history, as well as the interactions of dwellers over hundreds years of human settlement [1,2]. Therefore, the visual appearance of a place carries distinct information that differentiates it from other places. For example, by relying on the city-informative cues in a photo, one can infer the geo-location

[3]. From the style of the window and balcony support of a street view image, one can recognize that it is in Paris [4]. Understanding the visual differences between places is of great significance to urban studies, urban design and tourism planning. However, comparing and quantifying the visual discrepancies between places, as well as faithfully identifying the visual features in the physical setting of a place that makes the place distinctive, have been challenging and are not yet solved [5]. Meanwhile, because the visual cues are always subtle [4], people's high-level knowledge of cultural difference and scene perception are usually required to perceive and distinguish places [6,7].

The literature advanced by architects, urban planners and geographers has been active in understanding the visual characteristics of scenes, places and cities. However, previous works have mainly relied on *in situ* surveys and interviews on the visual identities in the urban space, such as exploring street elements that constitute people's mental maps [1], identifying place-identity symbols lost in a hurricane [8], examining place attachment at different scales [9] and exploring visual cues for place perception [10]. These methods provide valuable insights, but are labour-intensive, time-consuming and difficult to scale up [7]. On the other hand, the progress of computer vision has allowed the accurate recognition of the street-level imagery. Efforts have been made to localize a photo to the place where it was taken [11,12], determine the visual cues that make a city look special [4], understand what makes London look beautiful, quiet and happy [10]. These works have demonstrated that street-level images hold great promise for delivering more research value in understanding the physical setting of a place. However, our ability to understand the urban scene, for example, the cultural, historical style of streetscapes, is still limited by the lack of computational tools to extract high-level representations of images [13].

Recent breakthroughs in deep learning [14] have presented the outstanding performance of deep convolutional neural network (DCNN) in various computer vision tasks, such as image classification [15,16], image object detection [17] and image scene recognition [18]. They have produced recognition results comparable to or even better than human performance in some of the tasks. The success of DCNN is mostly attributed to its powerful ability to learn effective and interpretable image features [19]. Enabled by the proliferation of deep learning, researchers have advanced the studies on looking deeper into the physical appearance of places. It is even possible to infer factors beyond the scene, such as the crime rate [20], human feelings [21] and demographics [22]. Indeed, DCNN takes inspiration from the human brain and is designed to mimic human cognitive functions [14]. Recent studies have also demonstrated that humans and computer vision-based models behave alike in perceiving their surroundings by just seeing a small glimpse of it in a single street-level image [23]. Similarly, researchers have successfully trained DCNNs to learn for complex cognitive tasks—such as long-range navigation in cities without a map [24,25]. Compared with traditional computer vision features, DCNN learns and grasps high-level cognitive information of images, including complex visual concepts in the scenes, potentially helping to better capture the cultural and historical styles of places in street-level images.

The goal of this study is to extract distinctive visual aspect of places by exploring place-informative scenes and objects. Similar works have been done in recognizing geo-locations [26–28], detecting landmarks [11,12,29–31], examining ambiance perceptions [32,33] and identifying urban identities [4,5,34]. The work closest to ours is [4,5]. Doersch *et al.* [4] proposed a framework based on discriminative clustering to discover the clusters of local image patches that make a city distinct. These patch clusters reflect very local cues about the urban properties, such as windows style or building textures, while our work focuses on higher-level concepts such as scenes and objects. Zhou *et al.* [5] only analysed distinct scenes across cities using a scene classification framework, while our work unifies the analysis of scenes and objects. With different aims, this work proposes a general framework to explore a place's visual cues, not only including landmarks but also containing historical architecture, religious sites, unique urban scenes, unusual natural landscapes and distinct objects. In detail, the framework formulates the style learning problem as a discriminative classification task and first trains a DCNN-based model to predict the place where a given photo comes from. Second, by ranking the model confidence of each image sample, which indicates how much a photo visually looks like it was taken in a specific place, we capture a collection of place-informative scenes. We go one step further to explore the place-informative objects in the scenes by first detecting and cropping objects in the photos and then conducting model training and sample ranking tasks with the same pipeline. Third, by looking into the misclassification rate of the classification task, a metric to measure the distinctiveness of places and the similarity between a pair of places is proposed in the framework.

To demonstrate the effectiveness of the proposed framework, a case study is conducted by using more than two million photos from Panoramio[1] dataset of 18 cities worldwide to identify the most

---

[1]https://www.panoramio.com/.

city-informative objects and scenes, and measure the distinctiveness of cities and the similarity between a pair of cities. The results show that Bangkok, Rome, Amsterdam, etc., are the most visually distinctive cities, and Bangkok–Singapore, Hong Kong–Singapore and London–Paris, look similar to each other visually. Additionally, various landmarks, historical architecture, religious sites, unique urban scenes and unusual natural landscapes of each city have been explored as the most city-informative scenes. In terms of the city-informative objects, we take the most common objects, vehicles, as an example, finding that taxis, police cars and ambulances have been detected as the most city-informative vehicles, which is inspiring and consistent with common sense.

This work makes a contribution to learning the visual features of places with a DCNN using massive geo-tagged images. The work will further help to understand and quantify the visual characteristics of places and has the potential to enable insight into what geo-tagged visual big data can achieve in formalizing the concept of place, understanding urban morphology and advising city decision-makers.

# 2. Framework for exploring place-informative scenes and objects

We propose a general data-driven framework to learn and compare the visual appearance of different places, and explore the place-informative scenes and objects. Figure 1 depicts the overview of the framework. The framework is composed of four parts, namely (*a*) image preprocessing and database building, (*b*) training classification model to recognize places, (*c*) identifying place-informative scenes and objects and (*d*) calculating place distinctiveness and similarity. Part (*a*) illustrates the image data type and necessary preprocessing for the building of a geo-tagged image database. The image database is then employed to train a DCNN-based classification model to recognize which place the images come from. The training process is presented in part (*b*). Parts (*c*) and (*d*) identify place-informative scenes/objects and calculate place distinctiveness/similarity respectively based on the outputs of the model that trained in part (*b*). Details of each part are described below.

## 2.1. Geo-tagged image database

In order to analyse and understand the visual knowledge of a place, we use a large number of images with geographical information to represent the visual environment of the place. There are three commonly employed data sources to describe a place: social media photos, street view imagery and customized image data. Social media photos from photo-sharing websites like Panoramio and Flickr are uploaded by tourists and photographers. These photos record the beautiful or unique scenes of a city and present the subjective view of a place [32]. By contrast, street view imagery, such as Google Street View describe a place in an objective manner [35]. The content and locations of street view imagery are largely limited by the accessibility of the locations for street view vehicles. Besides, researchers will collect their own dataset by field survey to investigate specific places, such as indoor buildings on a campus [36].

The raw downloaded data is first preprocessed through data cleaning and data labelling. Data cleaning refers to the process of filtering the non-place-relevant data samples, especially for social media images which may contain artificial images, indoor scenes, objects, etc. Data labelling is to aggregate each image spatially to its place category. For instance, in order to examine the visual information across different cities, the images should be labelled by their city names. For $N$ places in the research area, we take all the image samples of the *i*th ($i = 0, 1, \ldots, N - 1$) place as the category label $i$.

The geo-tagged image database is then split into a training set and an application set with a ratio of $\phi$. The training set is used to train a classification model for learning the visual features and characteristics across places, and we explore the visually distinct scenes or objects by applying the classification model to the application set.

## 2.2. Place classification using deep convolutional neural network

People can easily identify a familiar place in a photo, because they can develop a knowledge of the place's characteristics through experience. Inspired by this cognitive capability, we formulate the place visual knowledge learning problem as an image discriminative classification task, aiming to predict which city a given image comes from by training a classifier. Our method assumes that the classifier will learn features and knowledge about visual differences among places.

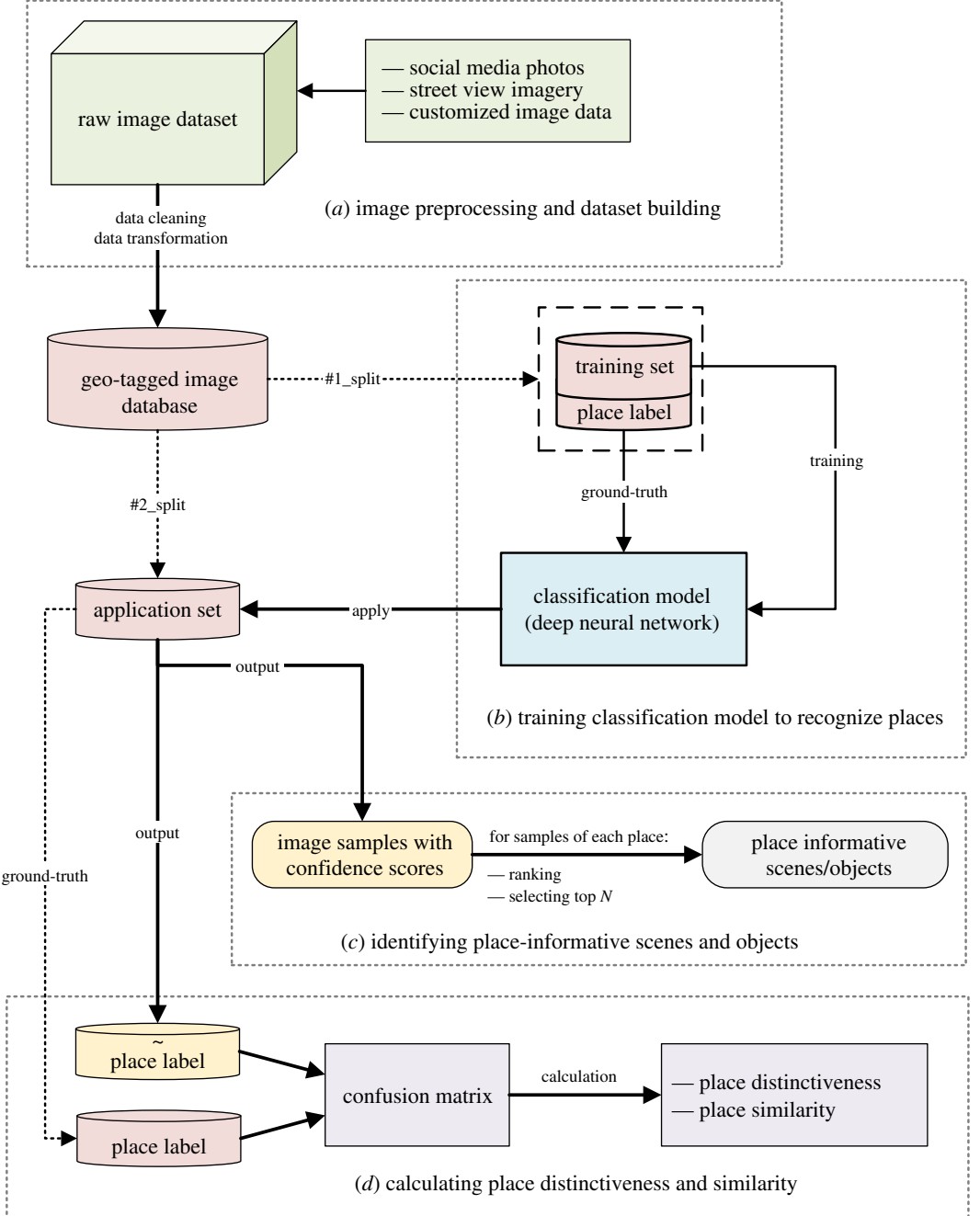

**Figure 1.** Framework for exploring place-informative scenes and objects. The framework is composed of four parts: (*a*) image preprocessing and database building; (*b*) training classification model to recognize places; (*c*) identifying place-informative scenes and objects; (*d*) calculating place distinctiveness and similarity.

The challenge of the classification task lies in the wide variety of scene contents, scene types and scene styles. Previous works using low/mid-level computer vision features have proved a certain performance for understanding scenes [26,37]. Recently, DCNN has been demonstrated as an effective tool to learn and grasp high-level visual knowledge of natural images, yielding state-of-the-art results compared with conventional features on various computer vision tasks [15,38]. The success of DCNN is mostly attributed to its powerful ability to learn effective and interpretable image features [19].

We suggest employing DCNN in this framework to better capture and learn the cultural and historical styles of places in the street-level imagery. For $N$ places in the research area, we train an $N$-category DCNN classification model using the training set. Since the sample numbers of places may be quite different, we assign weights to each categories to deal with the imbalanced class problem. For the $i$th place with $s$ image samples, the category weight of the place is obtained by $\bar{S}/s_i$, where $\bar{S}$ is the average number of the total samples.

We take advantage of the visual knowledge gained from the place image classification task and apply the pre-trained model to the application set. From the results, the confidence scores of image samples will be used to explore the most place-informative scenes and objects; and the misclassification will be used to measure the visual distinctiveness and similarity between places.

## 2.3. Identifying place-informative scenes and objects

To identify place-informative scenes and objects, we pose the problem with two objectives: (i) frequent itemset mining, which aims to identify the repeated and visually consistent samples in a same place category and further (ii) classification, i.e. mining samples occurring in a particular category rather than other categories. The two objectives can be typically approached by using machine learning-based classification. As described in step (b) of the framework, the pre-trained DCNN model is used to evaluate all the image samples in the application set.

Our approach is based on the confidence measure in the classification task. The assumption is that for a specific sample, the higher the probability that the model yields, the more distinctive the sample is in terms of discrimination from the other categories. We then rank the samples that would be correctly classified based on their confidence scores. The confidence score suggests how confident the algorithm is for a prediction [39]. In this case, it indicates how much a photo visually looks like it comes from a specific place. Through the process described above, a collection of city-informative scenes can be captured.

In terms of identifying the place-informative objects, we first organize a geo-tagged object image database that only contains object images. This process can be achieved by cropping object patches from social media photos using a pre-trained object detection model. Then, the place-informative objects can be explored through the same pipeline used in identifying the place-informative scenes (figure 2).

## 2.4. Visual distinctiveness and similarity between places

The misclassification rate of each place category is obtained from the classification task on the application set, and is then transformed into a confusion matrix. The confusion matrix is a metric that has been typically used to not only evaluate the performance of classification models on datasets, but also demonstrate the similarity between categories [40]. The values in the main diagonal represent the ratio for which the prediction is successful, while the other elements in the matrix are those that were mis-classified by the model. In this case, the diagonal values are actually the accuracy of the classification task for each place, and here, we believe it indicates the distinctiveness of each place, as a higher value indicates a larger number of distinct samples in the category that are not likely to be misclassified by the model. Hence, we take the diagonal values in the normalized confusion matrix as a metric to measure the visual distinctiveness of a place.

Additionally, the off-diagonal values indicate the misclassification ratio from place $P_i$ to place $P_j$. Here, we argue that if $P_i$ has a high probability to be recognized as $P_j$ and, at the same time $P_j$ has a high probability to be recognized as $P_i$, then the two places are visually similar to each other. Accordingly, the similarity score between $P_i$ and $P_j$ can be measured by the sum of the misclassification ratio of $P_i$ to $P_j$ and the misclassification ratio of $P_j$ to $P_i$, and we take the similarity score as the metric to measure the visual similarity between a place pair.

# 3. Experiment and results

In this section, we report the implementation of the framework by a case study. We train a DCNN with millions of social media photos to learn, compare and quantify the visual features between 18 cities worldwide.

## 3.1. Image data

The image data were collected from Panoramio. Panoramio is a photo-sharing platform that contains billions of geo-tagged photos taken from all around the world. It provides application programming interface (API) for photo data request, and all the photos used in this study were published under one of the Creative Commons licenses and can be used freely for academic purposes. In this study, we use

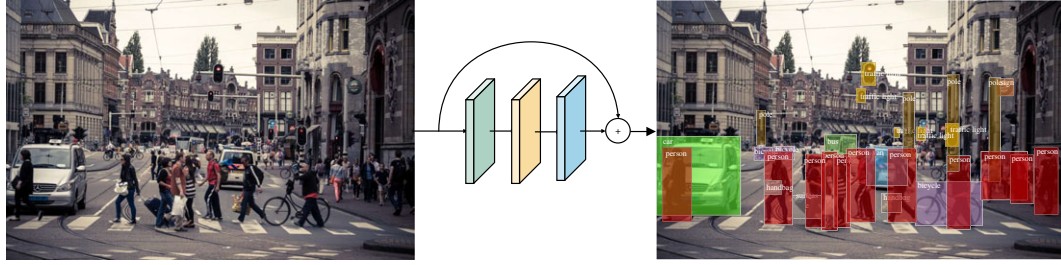

image object detection with deep convolutional neural network

**Figure 2.** Cropping object patch using DCNN. Photo credits: Panoramio dataset. Thanks for licensing this CC-BY.

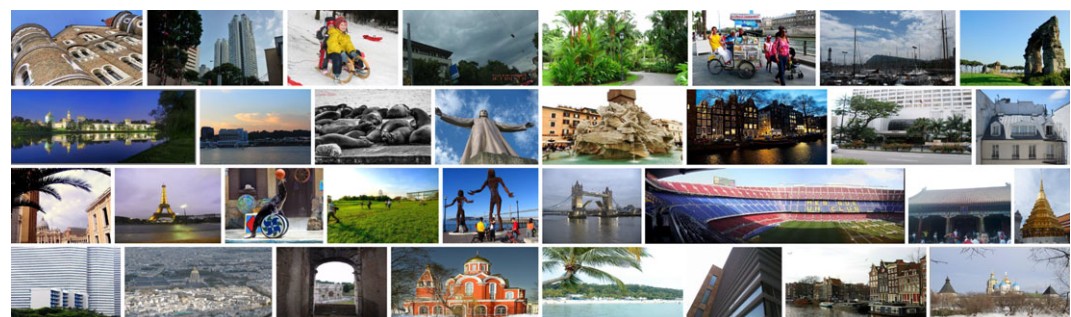

**Figure 3.** Photo samples in Panoramio dataset. Photo credits: Panoramio dataset. Thanks for licensing this CC-BY.

more than 2 million photos of Panoramio from 18 cities, including both metropolitan and small cities worldwide, spanning 15 countries across three continents. The photos were taken from 2013 to 2014 and uploaded by users, which include residents, photographers, tourists, etc., the image contents actually indicate their subjective preference of the cityscape, such as beautiful scenery, historic buildings, famous sculptures and ancient architecture, among other things.

We note that the photo types are of great variety, not only including outdoor scenes with different scales but also containing indoor scenes, objects (e.g. an apple) and artificial images (e.g. a logo). In order to remove these non-place-relevant images, we trained a binary-classification model to predict if an image is an outdoor scene image or not. We organized the training dataset with image samples from Caltech 101 (a dataset contains object images) [41] and Places2 [18] (a dataset contains indoor scenes and outdoor scenes), trained a support vector machine (SVM) classifier with Places2 image features [18] and achieved an accuracy of 97.94%. By filtering the initial dataset with the classifier, 2 084 117 photos were obtained for the experiments. Figure 3 presents several photo samples.

The geo-tagged city image database for learning city-informative scenes is then built by labelling all photo samples based on the cities they come from. To explore the city-informative objects, we organize a geo-tagged city object image database with object patches detected and cropped from each photo using a pre-trained image object detection model. In this case, a single-shot multibox detector (SSD) model [42] that is pre-trained on the common objects in context (COCO) dataset [43] is used. The pre-trained model is able to detect 80 categories of objects with an average accuracy of 81.6% [42]. The statistics of the photos and object patches are shown in table 1. The two databases were split into training set and application set with a ratio $\phi$ equal to 1 : 1.

## 3.2. Experiment

According to the proposed framework, the work flow of the experiment is divided into three steps. As depicted in figure 4, the first step is to conduct a city discriminative classification task, which predicts which city one given image is taken from. The misclassification rate of each city category is further analysed in step 2, in which we measure the visual similarity and visual distinctiveness with a confusion matrix. Additionally, the model trained in step 1 is further used to predict the confidence score of all the photo samples in step 3 to identify the city-informative scenes and objects of each city.

For the 18 cities, two 18-category classification DCNN models were trained using geo-tagged city image database and geo-tagged city object image database, respectively. In detail, we employed

**Table 1.** Statistics of the photo and photo patch data.

| sources | | | # images | # image patches | | | | | |
| city | country | continent | # photos | # car | # person | # truck | # bus | # boat | # train |
|---|---|---|---|---|---|---|---|---|---|
| Moscow | Russia | Europe | 291 371 | 51 386 | 41 080 | 4657 | 2573 | 2286 | 4982 |
| Tokyo | Japan | Asia | 247 260 | 26 959 | 33 430 | 3065 | 1616 | 1527 | 1362 |
| London | UK | Europe | 209 264 | 22 535 | 45 992 | 2040 | 4880 | 4383 | 2717 |
| New York | USA | America | 159 393 | 18 878 | 21 662 | 1635 | 1084 | 1939 | 3232 |
| Paris | France | Europe | 154 437 | 15 390 | 34 688 | 1618 | 1597 | 1426 | 4753 |
| Hong Kong | China | Asia | 152 147 | 14 150 | 18 793 | 1048 | 1583 | 355 | 681 |
| Berlin | Germany | Asia | 148 119 | 13 872 | 19 415 | 1400 | 1617 | 1340 | 996 |
| Barcelona | Spain | Europe | 114 867 | 11 145 | 12 318 | 1006 | 1396 | 1556 | 1110 |
| Bangkok | Thailand | Asia | 100 808 | 10 164 | 20 568 | 601 | 648 | 408 | 674 |
| Rome | Italy | Europe | 97 578 | 9861 | 12 695 | 533 | 1097 | 265 | 658 |
| Vienna | Austria | Europe | 89 380 | 9592 | 27 048 | 1675 | 4005 | 3979 | 1544 |
| Seoul | Korea | Asia | 89 007 | 8688 | 17 763 | 692 | 593 | 967 | 972 |
| Prague | Czech Republic | Europe | 74 984 | 8 086 | 21 922 | 737 | 887 | 1539 | 1050 |
| Amsterdam | The Netherlands | Europe | 67 853 | 7739 | 12 930 | 476 | 510 | 429 | 1273 |
| Singapore | Singapore | Asia | 66 364 | 7300 | 10 162 | 399 | 426 | 3821 | 667 |
| Beijing | China | Asia | 64 631 | 7248 | 10 531 | 565 | 599 | 587 | 1077 |
| San Francisco | USA | America | 64 592 | 6839 | 7190 | 605 | 629 | 812 | 567 |
| Toronto | Canada | America | 58 125 | 6108 | 11 113 | 754 | 649 | 1053 | 604 |

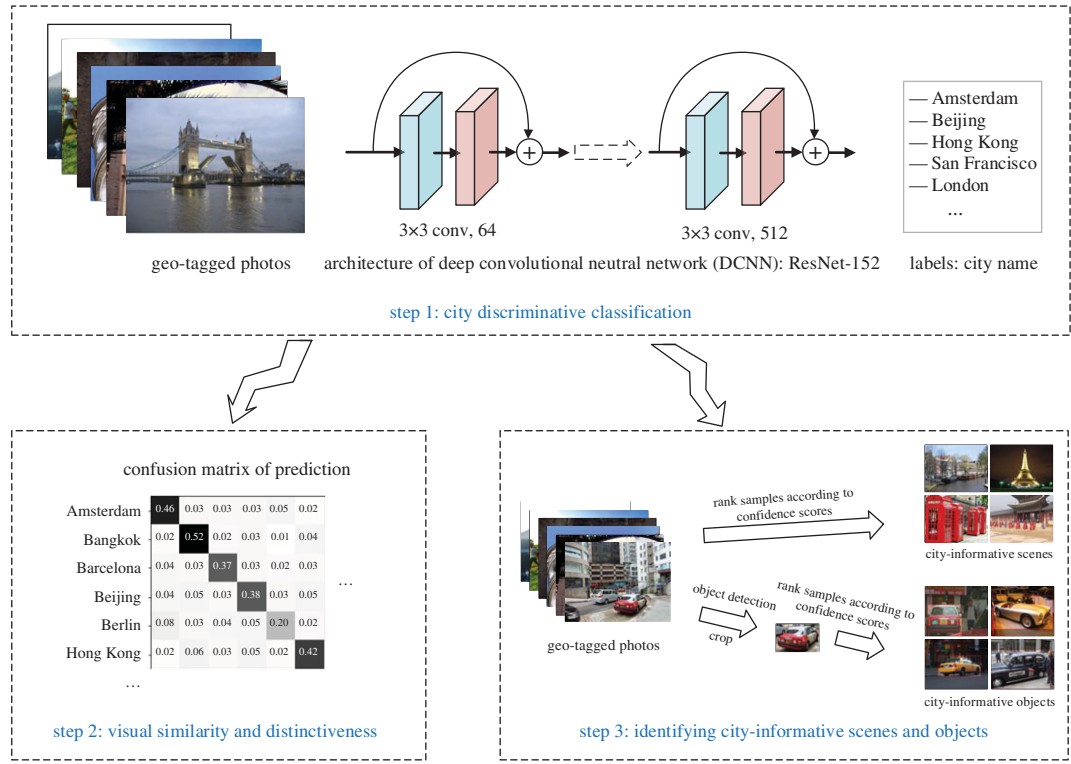

**Figure 4.** Overview of the experiment. The work flow is divided into three steps. In step 1, a city discriminative classification task is conducted to predict which city one given image come from. The misclassification rate of each city category is further analysed in step 2 for comparing the visual similarity and distinctiveness between cities. The model is used to predict the confidence scores of all the photo samples in step 3 for identifying city-informative scenes and objects. Photo credits: Panoramio dataset. Thanks for licensing this CC-BY.

ResNet-152 as the architecture of the model, which is a 152-layer DCNN model built by deep residual modules [44]. To accelerate the training process, the network was initialized with ImageNet weights [15]. In addition, we noted that the image numbers of each city are quite different. To deal with the imbalanced class problem, we applied weights to each city category.

For the city classification task using original photos, we achieved a 36.43% average accuracy over fivefold cross-validation with a $\pm 0.39\%$ confidential interval. For the city classification task using object patches, we achieved an accuracy of 50.15% for car, 38.61% for person, 45.31% for bus, 37.68% for truck, 32.29% for train and 32.29% for boat.

## 3.3. Results

### 3.3.1. Visual distinctiveness and similarity between cities

Based on the misclassification rate of the city classification task, we demonstrate the visual distinctiveness and similarity using the confusion matrix metric proposed in the framework. Figure 5 shows the confusion matrix. The main diagonal value represents the ratio of photo samples that are correctly classified into their city category, which indicates the visual distinctiveness of the city. The off-diagonal value represents the ratio of photo samples misclassified into another city category, which indicates the visual similarity between a pair of cities. A higher value indicates a higher visual similarity. Bangkok (0.52), Rome (0.50), Amsterdam (0.46), Seoul (0.44) and Hong Kong (0.42) are interpreted as more visually distinct, while Berlin (0.20), London (0.23) and Vienna (0.24) are easily misclassified into other city categories and are interpreted to be visually common.

In figure 6, we show the similarity matrix of the 18 cities. The similarity score between two cities is the sum of misclassification ratios of the pair of cities to each other. From the matrix, we concluded that Bangkok−Singapore (0.19), Beijing−San Francisco (0.17), Hong Kong−Singapore (0.16), London−Paris (0.15), Tokyo−San Francisco (0.15), Prague−Vienna (0.14) and Rome−Barcelona (0.14) were pairs of higher similarity.

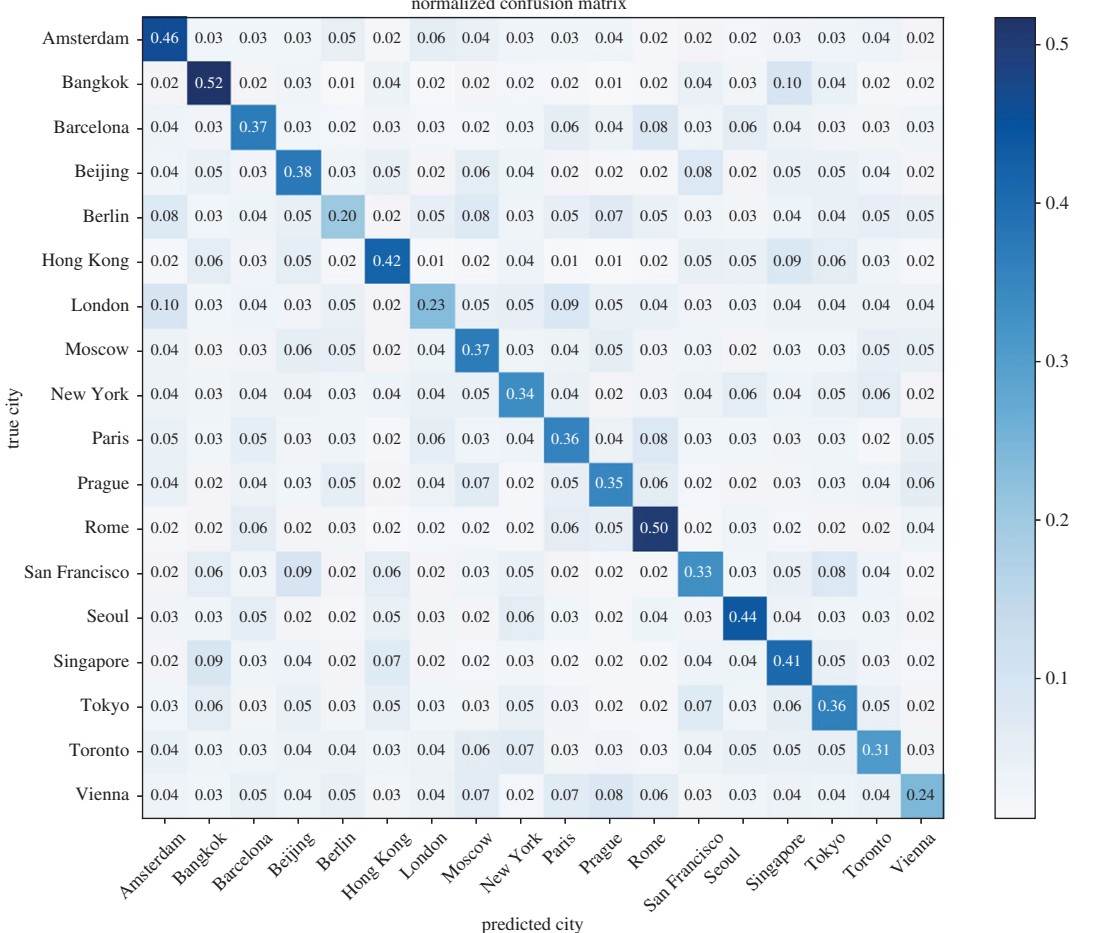

**Figure 5.** Confusion matrix of the city classification task using photos. The main diagonal values represent the ratio of photo samples that are correctly classified into their city category. The off-diagonal values represent the ratio of photo samples misclassified into another city category.

Figure 7 presents the geo-location of the 18 cities with red circles in a map. A larger circle indicates a higher distinctiveness score; city pairs with higher similarity scores are connected with blue lines. Generally, we noted that the cities geographically close to each other tended to be more visually similar, because of the cultural convergence and common history and origins, which is in accordance with common sense. Inspired by the two analyses above, we believed that there are city-informative cues that exist in the imagery, which enable the DCNN model to visually identify one city among others. In the next part, we present the results of mining these cues, which are city-informative scenes and objects.

### 3.3.2. Identifying city-informative scenes and objects

The DCNN model trained on the 18-city recognition task has learned the knowledge of cities' visual characteristics. We then used the model to predict the confidence score, which is the probability of each prediction being correct (ranging from 0 to 1) for all the photos, and ranked them from high to low. A higher confidence indicates that the model is more confident that one photo was taken from one specific city.

Figure 8 shows the photos with high confidence scores for each of the 18 cities, and these photos are believed to be the city-informative scenes. Landmarks, historical architecture, religious sites, unique urban scenes and unusual natural landscapes of the cities, which are city-informative and representative, have been explored according to their ranking. In particular, Amsterdam, Barcelona, Berlin, Hong Kong, Moscow, Prague and Vienna present their distinctive architectural styles; Beijing, Rome, Seoul and Tokyo show their identical and historical buildings; and London, New York, Paris and San Francisco are famous for several landmarks, such as the Eiffel Tower and the Golden Gate Bridge. Additionally, we notice that the cityscape characteristics and street patterns are greatly different among the cities of Singapore, Hong Kong, Prague and Tokyo.

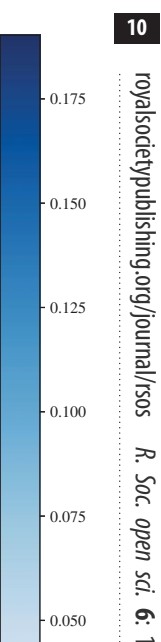

**Figure 6.** Visual similarity matrix of the 18 cities. The scores in the similarity matrix are calculated based on the confusion matrix, where the values and symmetry position along the diagonal were added up.

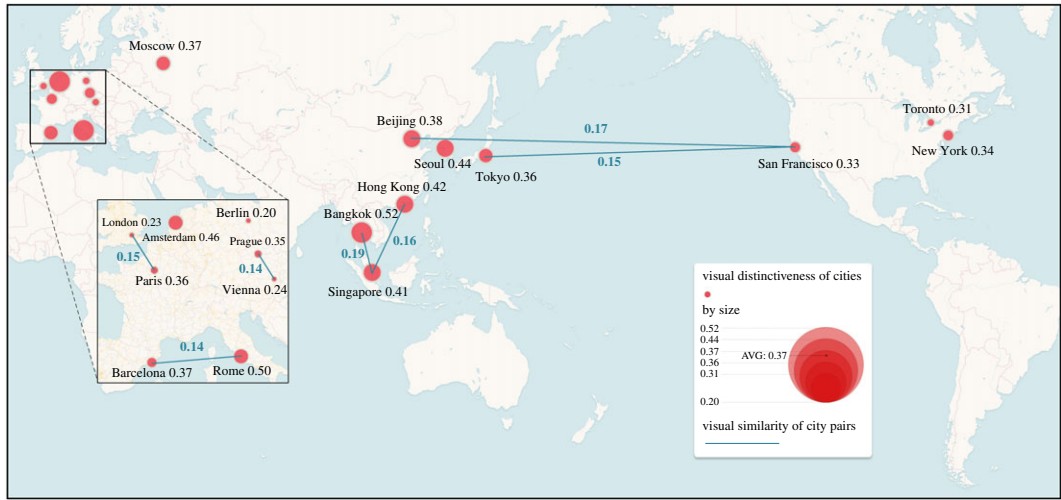

**Figure 7.** Geo-location of the 18 cities (red circles). A larger circle indicates higher distinctiveness score; city pairs with higher similarity scores are connected with blue lines.

Similarly, following the same pipeline, we used image patches, which show only particular objects in the photos, as the training samples for the 18-city classification task. In this case, we took the car as an example to explore the most city-informative cars. Figure 9 presents the most 'confident' car samples for each of the 18 cities. Intuitively, vehicles around the world should be visually similar, because car manufacturing is a global industry. Interestingly, as city-informative urban public transportation,

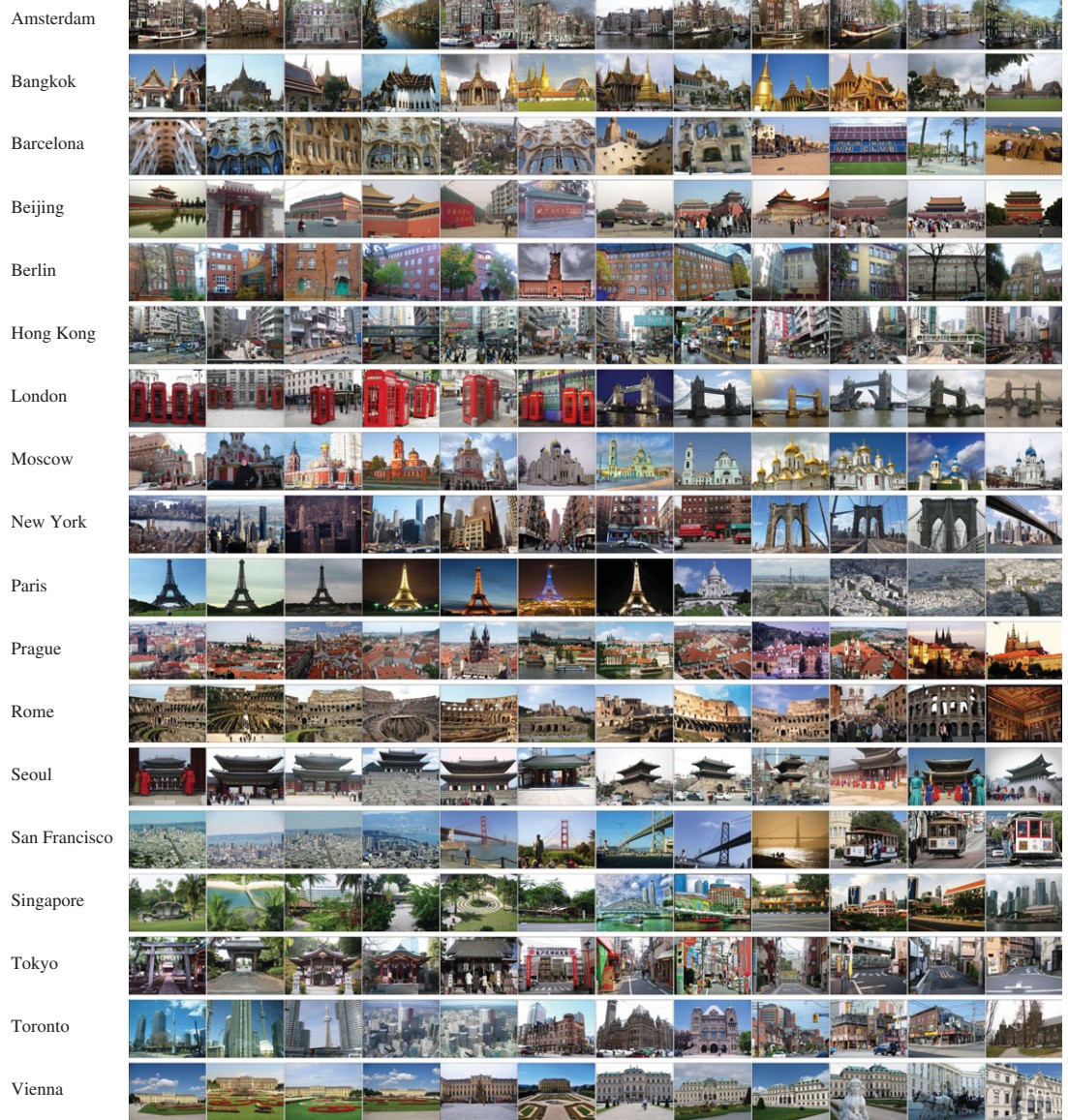

**Figure 8.** Samples of city-informative scenes for the 18 cities. Photo credits: Panoramio dataset. Thanks for licensing this CC-BY.

taxis, police cars and ambulances have been detected, which is consist with common sense because their appearances are different from each other among cities. Specifically, in Bangkok, Barcelona, Beijing, Hong Kong, London, New York, Singapore, Tokyo and Toronto, the taxis have been explored and identified significantly. Among them, for Bangkok, Hong Kong and Tokyo, more than one type of taxi has been detected. Additionally, we noted that people's preference for vehicle types are indeed different in the same cities. Cars in London are mostly retro-style and old-fashioned. In Paris, there are many racing and luxury cars. In Rome, we found large volumes of tiny cars, and in Moscow, the cars were always covered by snow.

## 4. Discussion

### 4.1. Understanding the personality and uniqueness of a city

The visual built environment reveals the personality and uniqueness of a place [1,45]. A single type of landmark and architecture cannot represent the urban identity comprehensively and the symbol of a city should be diverse and various to make it attracting, active and vital [1,46]. Previous studies attempted to understand the city uniqueness from the perspective of street details [4], landmarks [47] and areas of interest [48].

**Figure 9.** Samples of city-informative objects (cars) for the 18 cities. Photo credits: Panoramio dataset. Thanks for licensing this CC-BY.

In our case study, various visual cues of the 18 cities have been explored, not just in terms of landmarks but also including historical architecture, religious sites, unique urban scenes, unusual natural landscapes and distinct objects. The results achieved are mostly attributed to the ability of DCNN in learning and representing high-level visual features of the images. We believe that our results are of importance both to those working in urban management and design—in giving a complete overview of the city features that are unique and distinct from others and develop the brand image of a city for city development and tourism management—and also to geographers in general, in throwing light on understanding how a place is formed and shaped for place formalization in the process of human−place interaction.

## 4.2. Street-level imagery serves as an ideal proxy to represent the locale of a place

Enabled by the proliferation of wireless communication and mobile networking, the coming big-data era has led to the production of a considerable amount of geo-tagged images, such as Google Street View (GSV) images and images from social media services, which have been blanketing the all-round landscape of the urban area, providing new opportunities to enrich place semantics [49,50].

From this study, we conclude that street-level imagery involves visual information beyond different types of objects, uncovering the culture, history, religion, status of development, etc., of places. With

appropriate computational tools, street-level imagery serves as an ideal proxy to represent the locale of places, to understand different types of places, such as natural landscapes, built environments and indoor spaces, and exhibits great potential for a variety of disciplines.

## 4.3. Deep learning learns knowledge about the visual characteristics of a place

The framework was implemented via a discriminative classification task using a DCNN model. From the results, we can see that, for instance, taxis with different angles of view or even just parts shown in the image can be identified by the model as the most city-informative scenes or objects, which has demonstrated the model's ability to learn invariant features of an object. This ability is essential for this work in mining city-informative objects. Additionally, unlike the behaviour of traditional computer vision features in object recognition tasks, which relies on consistent features of an object, such as shape and colour, the DCNN-based model learns various stylistic information. For example, scenes including Tower Bridge or red telephone boxes are believed to be the most 'London style' for the model, but these two scenes have no common features. To some extent, we consider that DCNN has learned concepts and high-level representations of cityscapes.

The method proposed in this study to learn the visual knowledge about places has been demonstrated at a global city level. Similarly, it is also possible to compare different cities in the same country in order to learn how urban planning policy can have an impact on the appearance of the modern city. For instance, the fact that China is facing a problem of 'thousands of cities show the same' has been discussed widely [51]. It is of great potential to validate this question or further evaluate the similarity of these cities by employing the presented method. Additionally, it is promising to apply and generalize this method to the study of different scales, such as neighbourhood scale and indoor scale, to enable more insight into urban design and architectural design.

## 4.4. Limitation of the proposed framework

The performance of the proposed framework depends largely on the representativeness of the data source used. Crowd-sourced data are always biased towards a group of people that generates the data. For instance, the photos in the Panoramio dataset used in this study are mainly contributed by tourists and photographers, who will have a different perspective of a city's visual feature from the local residents. A single data source is therefore difficult describing the objective visual aspects of a city. Future studies are expected to integrate multiple data sources, e.g. Google Street View imagery, to represent the place comprehensively.

## 5. Conclusion

Formalizing the visual features, salient scenes and unique styles of places is crucial to urban design and urban management. This study proposes an image data-driven framework to learn visual knowledge of places and explore place-informative scenes and objects comprehensively, which can be applied to places of different types and scales, such as inter-city level, neighbourhood level and indoor level. The framework will provide the opportunities to the research of place formalization and place semantics.

In the case study, we investigated 18 cities worldwide with more than 2 million social media photos and explored unique visual features of each city including landmarks, historical architecture, religious sites, unique urban scenes, unusual natural landscapes, etc. In future works, we will look deeper into the spatial distribution of these unique visual features of different types for each city, and seek connections between visual features and the culture, geography and historical evolution of a city. The results of the case study support urban design practices and illustrate the value of employing machine learning methods to understand how cities are developed and formed uniquely.

This study also demonstrates the value of street-level imagery in representing the visual built environment of a place and the outstanding performance of DCNN in learning and representing the concepts and high-level visual knowledge of street-level imagery. The cultural, historical, religious and development status of a place can be potential uncovered through street-level imagery by employing DCNN.

Data accessibility. No data were generated from the work and all supporting data were obtained from previously published work available via the references below.

Authors' contributions. B.Z., F.Z. and C.R. conceived the experiment(s), F.Z. conducted the experiment(s); Y.L. and F.Z. analysed the results. All authors reviewed the manuscript and gave final approval for publication.

Competing interests. The authors declare no competing interests.

Funding. This work was supported by the National Key R&D Program of China under grant no. 2017YFB0503602, the National Natural Science Foundation of China under grant nos. 41830645 and 41625003, and China Postdoctoral Science Foundation under grant no. 2018M641068.

Acknowledgements. The authors thank Liu Liu of CitoryTech for collecting the image data used in this study, Allianz, Amsterdam Institute for Advanced Metropolitan Solutions, Brose, Cisco, Ericsson, Fraunhofer Institute, Liberty Mutual Institute, Kuwait-MIT Center for Natural Resources and the Environment, Shenzhen, Singapore-MIT Alliance for Research and Technology (SMART), UBER, Victoria State Government, Volkswagen Group America and all the members of the MIT Senseable City Lab Consortium for supporting this research.

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
