## [Reviewer comments · Royal Society Open Science]

Review History

RSOS-181375.R0 (Original submission)

Review form: Reviewer 1

Is the manuscript scientifically sound in its present form?

Yes

Are the interpretations and conclusions justified by the results?

No

Is the language acceptable?

Yes

Is it clear how to access all supporting data?

No

Do you have any ethical concerns with this paper?

No

Have you any concerns about statistical analyses in this paper?

No

Recommendation?

Reject

Comments to the Author(s)

The paper proposes a deep learning framework to explore the visual cues of cityscapes and to measure their distinctiveness. The author gather 2M photos from Panoramio and label them with the name of the city they have been taken in (for a total of 18 cities). They then attempt a prediction experiment by training a DCNN (initialized on ImageNet, as it is commonly done) to predict the label of the picture from visual cues. The accuracy is ~36%, over a random baseline of ~5%. What the authors focus on is to find out which are the cities with more distinctive visual patterns and to study the pairs of cities that are misclassified (A is classified as B and vice-versa). They also extract images and objects that are most distinctive of a given city.

The paper is nicely written and well-grounded in the existing literature of urban science, multimedia analysis and computer vision. However, the paper has some major limitations:

- 1) The contribution is quite minor, both in terms of technical advancements and in terms of findings. The framework proposed is quite standard: classification of pictures using well-established CNNs for classification and object detection combined with a straightforward error analysis. I don't think there's any ground for claiming that the main contribution is "a general framework to explore the visual cues of a place". The framework used existed already and the authors simply apply it to a specific use case.
- 2) The specific use case (or, in other terms, the main idea of identifying the distinctive visual cues of an urban space) is not novel either. Doersch et al. (correctly cited by the authors) already explored that idea and, even without using deep-learning (which at the time was not yet a popular and widely available tool) were able to get results that are much more granular and meaningful than the ones presented here: for example, knowing that a certain type of balcony gives to Paris a distinctive vibe is much more meaningful than saying that the Eiffel tower is visually distinctive of Paris. Many others built already on top of that work by exploring the ambiance of cities and places (indoor and outdoor), among other things -- see the work of Santani et al., for example (<https://dl.acm.org/citation.cfm?id=2806277>, <https://dl.acm.org/citation.cfm?id=3123402>, <https://dl.acm.org/citation.cfm?id=2967261>).
- 3) The way in which the methodology is applied leads to results which are quite expected and, in my opinion, are not interpreted correctly by the authors. Specifically, the authors are learning the distinctiveness of the visual patterns of a city by looking at the panoramio photos of that city. This type of dataset is by definition skewed towards few visual objects and landmarks: those that are already selected by the crowd as the most interesting (/distinctive) ones. In other words, panoramio is representative of the popular hotspots in the city. Given this training set, the network is mainly learning to classify hotspots, it is not learning the visual distinctiveness of the city as a whole. The results in Figure 5 could be obtained by simply showing representative pictures of clusters of similar images within the city (using a simple visual similarity metric), without using deep learning and without comparing cities between each other. In Paris, one would likely obtain a cluster for the Eiffel Tower and another for Montmartre. This is because in every city the crowd tends already to capture the most distinctive objects (e.g., red phone booths in London). So, in the end what this framework does is mainly to surface and compare landmarks in different cities. An alternative approach would be to use Google street view images instead. By doing so, the work would resemble the one of Doersch et al. but with a fancier deep learning framework on top, which would be still quite incremental, in my view.

A few other minor comments:

- "By filtering the initial dataset with a classifier" -> the authors should provide more details on how this is done. Filtering out some categories of images can skew the findings on image diversity even further.
- "we perform undersampling of the class with more samples" -> Toronto is the city with the fewest pictures (58k). Does this mean that the authors are using only 58k pictures per city? This number probably grows because of cross validation, but it would be fair to report the total number of pictures that have been used for the classification experiment (which is probably less than 2M).
- Check typos: \hat{A}^T vehicles

In short, I believe that, although the paper reads nicely and the experiments yield results that are intuitively meaningful, the novelty of the approach is insufficient and the results presented are just as good as listing popular objects in different locations and comparing them, which is OK but hardly novel or surprising.

Review form: Reviewer 2

Is the manuscript scientifically sound in its present form?

Yes

Are the interpretations and conclusions justified by the results?

Yes

Is the language acceptable?

Yes

Is it clear how to access all supporting data?

Not Applicable

Do you have any ethical concerns with this paper?

No

Have you any concerns about statistical analyses in this paper?

I do not feel qualified to assess the statistics

Recommendation?

Accept with minor revision (please list in comments)

Comments to the Author(s)

This paper presents results of the study which refers to training a DCNN model with millions of images from Panoramio. The goal of this study is to learn the visual knowledge about cities and made an effort to explore the city-informative scenes and objects in a data-driven and machine learning manner.

The topic of the paper is very interesting and actual. The paper is technically correct and with content relevant to the journal readers. Results of this study are scientifically based and well presented.

The first part of article is very well-written and clearly states the problem, covers the various related works.

At the end of the paper, the authors listed significance in several aspects of their research results. The approach the authors presented is very complex. Obviously, a lot of effort has been invested in the implementation of the proposed solution.

But, because of the complexity, many details of the proposed solution are not seen.

As authors noted in Introduction section, the goal of this study is to build a data-driven framework to learn and compare the visual appearance of different cities. However, there are no details about this framework. The paper describes only the method, i.e. work flow divided into three steps.

Decision letter (RSOS-181375.R0)

24-Oct-2018

Dear Dr Zhang,

The editors assigned to your paper ("Exploring place-informative scenes and objects") have now received comments from reviewers. We would like you to revise your paper in accordance with the referee and Associate Editor suggestions which can be found below (not including confidential reports to the Editor). Please note this decision does not guarantee eventual acceptance.

Please submit a copy of your revised paper before 16-Nov-2018. Please note that the revision deadline will expire at 00.00am on this date. If we do not hear from you within this time then it will be assumed that the paper has been withdrawn. In exceptional circumstances, extensions may be possible if agreed with the Editorial Office in advance. We do not allow multiple rounds of revision so we urge you to make every effort to fully address all of the comments at this stage. If deemed necessary by the Editors, your manuscript will be sent back to one or more of the original reviewers for assessment. If the original reviewers are not available, we may invite new reviewers.

- Data accessibility

If you wish to submit your supporting data or code to Dryad (<http://datadryad.org/>), or modify your current submission to dryad, please use the following link:
<http://datadryad.org/submit?journalID=RSOS&manu=RSOS-181375>

- Competing interests

- Authors' contributions

- Acknowledgements

- Funding statement

Please note that Royal Society Open Science charge article processing charges for all new submissions that are accepted for publication. Charges will also apply to papers transferred to Royal Society Open Science from other Royal Society Publishing journals, as well as papers submitted as part of our collaboration with the Royal Society of Chemistry (<http://rsos.royalsocietypublishing.org/chemistry>). If your manuscript is newly submitted and subsequently accepted for publication, you will be asked to pay the article processing charge, unless you request a waiver and this is approved by Royal Society Publishing. You can find out more about the charges at <http://rsos.royalsocietypublishing.org/page/charges>. Should you have any queries, please contact openscience@royalsociety.org.

on behalf of Dr Cecilia Mascolo (Associate Editor) and Prof. Marta Kwiatkowska (Subject Editor)
openscience@royalsociety.org

Associate Editor's comments (Dr Cecilia Mascolo):

Both reviewers are generally lacking descriptions on the technicalities of the approach. One reviewer highlights the lack of novelty with respect to a number of works and challenges the interpretation of the results. I would like to give the authors a chance to answer these comments before making a final decision on the paper.

Comments to Author:

Reviewers' Comments to Author:

Reviewer: 1

Comments to the Author(s)

The paper proposes a deep learning framework to explore the visual cues of cityscapes and to measure their distinctiveness. The author gather 2M photos from Panoramio and label them with the name of the city they have been taken in (for a total of 18 cities). They then attempt a prediction experiment by training a DCNN (initialized on ImageNet, as it is commonly done) to predict the label of the picture from visual cues. The accuracy is ~36%, over a random baseline of ~5%. What the authors focus on is to find out which are the cities with more distinctive visual patterns and to study the pairs of cities that are misclassified (A is classified as B and vice-versa). They also extract images and objects that are most distinctive of a given city.

The paper is nicely written and well-grounded in the existing literature of urban science, multimedia analysis and computer vision. However, the paper has some major limitations:

1) The contribution is quite minor, both in terms of technical advancements and in terms of findings. The framework proposed is quite standard: classification of pictures using well-established CNNs for classification and object detection combined with a straightforward error analysis. I don't think there's any ground for claiming that the main contribution is "a general framework to explore the visual cues of a place". The framework used existed already and the authors simply apply it to a specific use case.

2) The specific use case (or, in other terms, the main idea of identifying the distinctive visual cues of an urban space) is not novel either. Doersch et al. (correctly cited by the authors) already explored that idea and, even without using deep-learning (which at the time was not yet a popular and widely available tool) were able to get results that are much more granular and meaningful than the ones presented here: for example, knowing that a certain type of balcony gives to Paris a distinctive vibe is much more meaningful than saying that the Eiffel tower is visually distinctive of Paris. Many others built already on top of that work by exploring the ambiance of cities and places (indoor and outdoor), among other things -- see the work of Santani

et al., for example (<https://dl.acm.org/citation.cfm?id=2806277>,
<https://dl.acm.org/citation.cfm?id=3123402>, <https://dl.acm.org/citation.cfm?id=2967261>).

3) The way in which the methodology is applied leads to results which are quite expected and, in my opinion, are not interpreted correctly by the authors. Specifically, the authors are learning the distinctiveness of the visual patterns of a city by looking at the panoramio photos of that city. This type of dataset is by definition skewed towards few visual objects and landmarks: those that are already selected by the crowd as the most interesting (/distinctive) ones. In other words, panoramio is representative of the popular hotspots in the city. Given this training set, the network is mainly learning to classify hotspots, it is not learning the visual distinctiveness of the city as a whole. The results in Figure 5 could be obtained by simply showing representative pictures of clusters of similar images within the city (using a simple visual similarity metric), without using deep learning and without comparing cities between each other. In Paris, one would likely obtain a cluster for the Eiffel Tower and another for Montmartre. This is because in every city the crowd tends already to capture the most distinctive objects (e.g., red phone boots in London). So, in the end what this framework does is mainly to surface and compare landmarks in different cities. An alternative approach would be to use Google street view images instead. By doing so, the work would resemble the one of Doersch et al. but with a fancier deep learning framework on top, which would be still quite incremental, in my view.

A few other minor comments:

- "By filtering the initial dataset with a classifier" -> the authors should provide more details on how this is done. Filtering out some categories of images can skew the findings on image diversity even further.
- "we perform undersampling of the class with more samples" -> Toronto is the city with the fewest pictures (58k). Does this mean that the authors are using only 58k pictures per city? This number probably grows because of cross validation, but it would be fair to report the total number of pictures that have been used for the classification experiment (which is probably less than 2M).
- Check typos: \hat{A} \hat{T} vehicles

In short, I believe that, although the paper reads nicely and the experiments yield results that are intuitively meaningful, the novelty of the approach is insufficient and the results presented are just as good as listing popular objects in different locations and comparing them, which is OK but hardly novel or surprising.

Reviewer: 2

Comments to the Author(s)

This paper presents results of the study which refers to training a DCNN model with millions of images from Panoramio. The goal of this study is to learn the visual knowledge about cities and made an effort to explore the city-informative scenes and objects in a data-driven and machine learning manner.

The topic of the paper is very interesting and actual. The paper is technically correct and with content relevant to the journal readers. Results of this study are scientifically based and well presented.

The first part of article is very well-written and clearly states the problem, covers the various related works.

At the end of the paper, the authors listed significance in several aspects of their research results. The approach the authors presented is very complex. Obviously, a lot of effort has been invested in the implementation of the proposed solution.

But, because of the complexity, many details of the proposed solution are not seen.

As authors noted in Introduction section, the goal of this study is to build a data-driven framework to learn and compare the visual appearance of different cities. However, there are no details about this framework. The paper describes only the method, i.e. work flow divided into three steps.

Author's Response to Decision Letter for (RSOS-181375.R0)

See Appendix A.

RSOS-181375.R1 (Revision)

Review form: Reviewer 1

Is the manuscript scientifically sound in its present form?

Yes

Are the interpretations and conclusions justified by the results?

Yes

Is the language acceptable?

Yes

Is it clear how to access all supporting data?

No

Do you have any ethical concerns with this paper?

No

Have you any concerns about statistical analyses in this paper?

No

Recommendation?

Accept with minor revision (please list in comments)

Comments to the Author(s)

I do acknowledge the effort of the authors in improving the paper. However, there is little remedy to add novelty to a contribution which has very little novelty. I am fine in seeing this paper accepted because it is technically correct but only if the authors agree on explicitly acknowledging the limitations of their contribution. This would be done with the following two actions:

1) The workflow is indeed better explained now, which is good. Yet, a better explanation doesn't make it any more novel. The authors simply don't make a methodological contribution in this space and this has to be acknowledged. Sentences like "This work makes a contribution to learning the visual features of places by proposing a DCNN and geo-tagged images based

framework" and "The goal of this study is to build a data-driven framework to learn and compare the visual appearance of different places by exploring place-informative scenes and objects" need to be either removed or rephrased. What the authors do is to ****use an existing visual processing framework to extract distinctive visual aspects of cities**** -- this has to be clearly stated.

2) I don't think it is sufficient to cite the work of Santani and Doersch. In the related work or in the introduction (or in any other relevant part of the manuscript) I would like to see one or two paragraphs discussing explicitly those works, acknowledging how this paper methodologically builds upon them and highlighting the differences.

I am sorry but I did not understand the purpose of the additional experiment reported. I would encourage the authors to make an additional effort to motivate it better.

Regarding the application on Panoramio photos vs. other sources: the application on Panoramio does not make the paper wrong but it limits its interestingness, in my opinion, which will be probably reflected in a reduced impact of the manuscript in the community. That said, I am fine with it as long as the potential limitations are mentioned.

Decision letter (RSOS-181375.R1)

25-Jan-2019

Dear Dr ZHANG:

On behalf of the Editors, I am pleased to inform you that your Manuscript RSOS-181375.R1 entitled "Exploring place-informative scenes and objects" has been accepted for publication in Royal Society Open Science subject to minor revision in accordance with the referee suggestions. Please find the referees' comments at the end of this email.

The reviewers and Subject Editor have recommended publication, but also suggest some minor revisions to your manuscript. Therefore, I invite you to respond to the comments and revise your manuscript.

- Ethics statement

- Data accessibility

If you wish to submit your supporting data or code to Dryad (<http://datadryad.org/>), or modify your current submission to dryad, please use the following link:
<http://datadryad.org/submit?journalID=RSOS&manu=RSOS-181375.R1>

- **Competing interests**

- **Authors' contributions**

- **Acknowledgements**

- **Funding statement**

Because the schedule for publication is very tight, it is a condition of publication that you submit the revised version of your manuscript before 03-Feb-2019. Please note that the revision deadline will expire at 00.00am on this date. If you do not think you will be able to meet this date please let me know immediately.

When submitting your revised manuscript, you will be able to respond to the comments made by the referees and upload a file "Response to Referees" in "Section 6 - File Upload". You can use this to document any changes you make to the original manuscript. In order to expedite the

processing of the revised manuscript, please be as specific as possible in your response to the referees.

Kind regards,
Andrew Dunn
Senior Publishing Editor
Royal Society Open Science
openscience@royalsociety.org

on behalf of Dr Cecilia Mascolo (Associate Editor) and Marta Kwiatkowska (Subject Editor)
openscience@royalsociety.org

Associate Editor Comments to Author (Dr Cecilia Mascolo):

There are still a number of presentation issue related to how the work is positioned and how the additional evaluation is described which need fixing. I will check that the authors have fixed this before the paper can be accepted.

Reviewer comments to Author:
Reviewer: 1

Comments to the Author(s)

I do acknowledge the effort of the authors in improving the paper. However, there is little remedy to add novelty to a contribution which has very little novelty. I am fine in seeing this

paper accepted because it is technically correct but only if the authors agree on explicitly acknowledging the limitations of their contribution. This would be done with the following two actions:

1) The workflow is indeed better explained now, which is good. Yet, a better explanation doesn't make it any more novel. The authors simply don't make a methodological contribution in this space and this has to be acknowledged. Sentences like "This work makes a contribution to learning the visual features of places by proposing a DCNN and geo-tagged images based framework" and "The goal of this study is to build a data-driven framework to learn and compare the visual appearance of different places by exploring place-informative scenes and objects" need to be either removed or rephrased. What the authors do is to **use an existing visual processing framework to extract distinctive visual aspects of cities** -- this has to be clearly stated.

2) I don't think it is sufficient to cite the work of Santani and Doersch. In the related work or in the introduction (or in any other relevant part of the manuscript) I would like to see one or two paragraphs discussing explicitly those works, acknowledging how this paper methodologically builds upon them and highlighting the differences.

I am sorry but I did not understand the purpose of the additional experiment reported. I would encourage the authors to make an additional effort to motivate it better.

Regarding the application on Panoramio photos vs. other sources: the application on Panoramio does not make the paper wrong but it limits its interestingness, in my opinion, which will be probably reflected in a reduced impact of the manuscript in the community. That said, I am fine with it as long as the potential limitations are mentioned.

Author's Response to Decision Letter for (RSOS-181375.R1)

See Appendix B.

Decision letter (RSOS-181375.R2)

07-Feb-2019

Dear Dr Zhang,

I am pleased to inform you that your manuscript entitled "Discovering place-informative scenes and objects" is now accepted for publication in Royal Society Open Science.

Royal Society Open Science operates under a continuous publication model

(<http://bit.ly/cpFAQ>). Your article will be published straight into the next open issue and this will be the final version of the paper. As such, it can be cited immediately by other researchers. As the issue version of your paper will be the only version to be published I would advise you to check your proofs thoroughly as changes cannot be made once the paper is published.

on behalf of Dr Cecilia Mascolo (Associate Editor) and Professor Marta Kwiatkowska (Subject Editor)
openscience@royalsociety.org

Appendix A

Reviewer 1

Comments to the Author(s)

The paper proposes a deep learning framework to explore the visual cues of cityscapes and to measure their distinctiveness. The author gather 2M photos from Panoramio and label them with the name of the city they have been taken in (for a total of 18 cities). They then attempt a prediction experiment by training a DCNN (initialized on ImageNet, as it is commonly done) to predict the label of the picture from visual cues. The accuracy is ~36%, over a random baseline of ~5%. What the authors focus on is to find out which are the cities with more distinctive visual patterns and to study the pairs of cities that are misclassified (A is classified as B and vice-versa). They also extract images and objects that are most distinctive of a given city.

The paper is nicely written and well-grounded in the existing literature of urban science, multimedia analysis and computer vision. However, the paper has some major limitations:

1) The contribution is quite minor, both in terms of technical advancements and in terms of findings. The framework proposed is quite standard: classification of pictures using well-established CNNs for classification and object detection combined with a straightforward error analysis. I don't think there's any ground for claiming that the main contribution is "a general framework to explore the visual cues of a place". The framework used existed already and the authors simply apply it to a specific use case.

Thank you very much for your constructive comments. We made major revision on the manuscript according to your comments and suggestions. In the revision, we restructured the whole manuscript, added a new section to introduce the framework of exploring place-informative objects and scenes, conducted a follow-up experiment to discuss the new findings about geo-bias in image database, and added a new discussion section to highlight the findings of this work. We believe that our work contributes to the following aspects:

1. We proposed a general data-drive framework to learn and compare the visual appearance of different places, and explore place-informative scenes and objects comprehensively, which can be applied to places of different types and scales, such as inter-city level, neighborhood-level, and indoor-level. A deep convolutional neural network is used in the framework to learn and represent the visual concepts and high-level visual knowledge of images, which enables the framework to explore a various types of uniqueness of a place. The framework will provide the opportunities to the research of place formalization and place semantics, which exhibits great potential for a variety of disciplines.
2. Understanding the uniqueness of a city comprehensively is important, and the focus on landmark as a single feature reduces the extent the research can be generalized. In the case study, we explored the city-informative scenes and objects over 18 cities. With the help of a DCNN model, we identify that other than landmarks, a large number of historical architecture, religious sites, unique urban scenes, and some unusual natural landscapes to be the most city-informative scenes. We believe that our results are of importance both to those working in the urban management and design - in giving an complete overview of the city features that are unique and distinct from others and develop the brand image of a city for city development and tourism management - and also to geographers in general,

in throwing light on understanding how a place is formed and shaped for place formalization in the process of human-place interaction.

3. A follow-up experiment was conducted the geo-bias existed in common image database. The experiment recognize cities using concatenated image features that extracted from scene context and scene object image separately. The accuracy of the city classification task was improve significantly compared to the baseline experiment. We conclude that the scene objects and scene context typically and complementary constitute a city-informative scene. They contain different city attribute information and accordingly contribute to the city recognition task separately. This finding has implications for improving the accuracy of object detection and recognition task in computer vision by taking scene context information into consideration.

The methods in this framework, for instance, ranking samples according to model's confidence score to list the most representative samples in the dataset, has been used in the field of machine learning and computer vision applications. Nevertheless, it is still worth a try to apply this idea to a more boarder field with a general interest, such as understanding the discrepancy, heterogeneity and diversity of different places with different scales, which exhibits great potential for a variety of disciplines. Hence, we propose this general framework to the field with interest of its application comprehensively.

2) The specific use case (or, in other terms, the main idea of identifying the distinctive visual cues of an urban space) is not novel either. Doersch et al. (correctly cited by the authors) already explored that idea and, even without using deep-learning (which at the time was not yet a popular and widely available tool) were able to get results that are much more granular and meaningful than the ones presented here: for example, knowing that a certain type of balcony gives to Paris a distinctive vibe is much more meaningful than saying that the Eiffel tower is visually distinctive of Paris. Many others built already on top of that work by exploring the ambiance of cities and places (indoor and outdoor), among other things -- see the work of Santani et al., for example (<https://dl.acm.org/citation.cfm?id=2806277>, <https://dl.acm.org/citation.cfm?id=3123402>, <https://dl.acm.org/citation.cfm?id=2967261>).

Thanks for pointing this out. Doersch et al. did an excellent work in identifying the representative visual cues of an urban space. We take his study as our major reference but we are of different objectives. Instead of detecting a landmark of cities and discovering local discriminative patches, our work aims at analyzing the city identity of various types. Actually, as the Figure (a) shown below, our model explored not only landmarks, but also historical architecture, religious sites, unique urban scenes, interesting objects, and unusual natural landscapes. A single type of landmark and architecture cannot represent the urban identity comprehensively and the symbol of a city should be diverse and various to make it attracting, active, and vital (Lynch, 1960; Jacobs, 1992). This result will give an urban designer or urban decision maker a more comprehensive picture to understand the uniqueness of a city compared to others. Doersch et al.'s work first obtained millions of image patches and employed HOG+color descriptor for clustering and classification. Their method has advantage in exploring details of architectures and buildings, remain difficulty in understanding the overall aesthetic, cultural and historical style of a cityscape or streetscape.

Thanks for suggesting the works from Santani et al., the method used and the problem identified are very inspiring to our study and we have added their works to this manuscript as our reference.

Figure (a). Historical architecture, religious sites, unique urban scenes, interesting objects, and unusual natural landscapes are identified in the experiment.

3) The way in which the methodology is applied leads to results which are quite expected and, in my opinion, are not interpreted correctly by the authors. Specifically, the authors are learning the distinctiveness of the visual patterns of a city by looking at the panoramic photos of that city. This type of dataset is by definition skewed towards few visual objects and landmarks: those that are already selected by the crowd as the most interesting (/distinctive) ones. In other words, panoramio is representative of the popular hotspots in the city. Given this training set, the network is mainly learning to classify hotspots, it is not learning the visual distinctiveness of the city as a whole. The results in Figure 5 could be obtained by simply showing representative pictures of clusters of similar images within the city (using a simple visual similarity metric), without using deep learning and without comparing cities between each other. In Paris, one

would likely obtain a cluster for the Eiffel Tower and another for Montmartre. This is because in every city the crowd tends already to capture the most distinctive objects (e.g., red phone booths in London). So, in the end what this framework does is mainly to surface and compare landmarks in different cities. An alternative approach would be to use Google street view images instead. By doing so, the work would resemble the one of Doersch et al. but with a fancier deep learning framework on top, which would be still quite incremental, in my view.

Thanks for your constructive comments and suggestions. We totally agree with you that the Panoramio dataset is skewed and biased, and the representative of this data source for the visual environment of a city is still need to be discussed, just like almost all of the crowdsourcing data.

From another perspective, Panoramio is an appropriate choice in terms of exploring the unique objects and scenes of a city. People took pictures because the scenes being in front of them are beautiful, unique, interesting, or different from other places. If people believe a scene to be boring and normal, they will rarely want to take a picture and upload to Panoramio. Hence, we believe that “biased or not biased” of a dataset depends on what research problems we want to address. We are agree with you that “Panoramio is representative of the popular hotspots in the city”, we can believe that the places in these hotspots are exactly the places we want to focus on, which contains the most unique scenes in the city. Nevertheless, we still need to focus on how to learn the concepts and visual features by a model to explore the real significant unique scenes and objects because there are a lot of noise in the social media dataset. Images from Google Street View are taken on roads where the Google vehicle can go, so the content of these images is limited. For instance, we would have difficulty in seeing the scenes shown in Figure (a) from GSV, so we can hardly observe the scenes like in our findings by using GSV data. In our manuscript, we have added a discussion on different data sources in the framework section. In conclusion, your suggestions are very valuable for us and we will consider using images both from the social media and street view images in our future study.

Doersch et al.’s work focuses on exploring the architectural details through local discriminative patches and proposes an ingenious workflow. Our study focuses on a different ‘scale’ of the uniqueness. It will be of significance to observing the spatial distribution of the uniqueness of different types - historical architecture, religious sites, unique urban scenes and unusual natural landscapes, which would help urban designers to understand the connections between visual features and the culture, geography, and historical evolutions of a city, and to develop the brand image of a city in terms of city development and tourism management.

In terms of clustering vs deep learning, we believe that Figure 5 is hard to be obtained by “clustering with a simple visual similarity metric without deep learning”.

On one hand, in Figure 5 we show not only landmarks, but also, historical architecture, religious sites, unique urban scenes, and unusual natural landscapes of each city. These scenes are hardly obtained by simply clustering the images with low/mid-level features, such as HOG+color descriptor used in Doersch et al.’s work, because the scenes are complex visual concept and of cultural and historical styles of places. A deep convolutional neural network is exactly used to model and learn the concepts and high-level representation of an image and even perform some of high-level cognitive tasks like human, which will make a difference in our case.

On the other hand, simply clustering will have difficult in handling the images of great diversity and noise (as Figure (b) shows). A great deal of images with similar low/mid level visual features would be clustered together, but they are not place-informative. A CNN is always compared with conventional low/mid-level image features, such as GIST, SIFT, and HOG several years ago. It is just like one paper you recommended (<https://dl.acm.org/citation.cfm?id=2967261>). Over these 2-

3 years, works in computer vision field compare the performance among different CNN models instead of conventional image features, because a CNN model will usually outperform them to a large extent.

Figure (b). Samples of the image data in the Panoramio dataset.

- "By filtering the initial dataset with a classifier" -> the authors should provide more details on how this is done. Filtering out some categories of images can skew the findings on image diversity even further.

Thanks for your comments. We have added more detailed descriptions on noise image filtering in the revision as your request. In our case, we filter the initial dataset with a DCNN based classifier. As the Figure (c) shows, we note that the photo types are of great variety, not only including the outdoor scenes with different scales but also containing indoor scenes, objects (e.g. an apple) and artificial images (e.g. a logo). In order to remove these non-place relevant images, we trained a binary-classification model to predict if an image to be an outdoor scene image or not. We organized the training dataset with image samples from Caltech 101 (a dataset contains object images) (Li et al. 2007) and Places2 (Zhou et al. 2017) (a dataset contains indoor scenes and outdoor scenes). We then trained an SVM classifier with Places2 image features and achieved an accuracy of 97.94%. By filtering the initial dataset with the classifier, 2,084,117 photos were obtained for the experiments.

Only indoor images, objects and artificial images are removed from the initial dataset, and we believe this operation will not skew the findings. Thanks for pointing this issue out, we have added this in the manuscript.

Figure (c). Samples of the noisy image data in the Panoramio dataset. There are indoor scene, objects and logo, which are non-urban images

A few other minor comments:

- "we perform undersampling of the class with more samples" -> Toronto is the city with the fewest pictures (58k). Does this mean that the authors are using only 58k pictures per city? This number probably grows because of cross validation, but it would be fair to report the total number of pictures that have been used for the classification experiment (which is probably less than 2M).

Thanks for your reminder. We made a mistake in describing this process. In our final trial of experiment, we apply weights to each categories to address the imbalanced issue. The weights can be calculated by: $\frac{\bar{N}}{N_i}$, where \bar{N} is the average number of samples of all cities, and N_i is the number of samples of the i^{th} ($i \in [1,18]$) city.

- Check typos: \hat{A} vehicles

Thanks for reminding. We have revised this mistake.

Reviewer 2

This paper presents results of the study which refers to training a DCNN model with millions of images from Panoramio. The goal of this study is to learn the visual knowledge about cities and made an effort to explore the city-informative scenes and objects in a data-driven and machine learning manner.

The topic of the paper is very interesting and actual. The paper is technically correct and with content relevant to the journal readers. Results of this study are scientifically based

and well presented.

The first part of article is very well-written and clearly states the problem, covers the various related works.

At the end of the paper, the authors listed significance in several aspects of their research results.

The approach the authors presented is very complex. Obviously, a lot of effort has been invested in the implementation of the proposed solution.

But, because of the complexity, many details of the proposed solution are not seen.

As authors noted in Introduction section, the goal of this study is to build a data-driven framework to learn and compare the visual appearance of different cities. However, there are no details about this framework. The paper describes only the method, i.e. work flow divided into three steps.

We appreciate the positive comments on our manuscript and thanks for these constructive suggestions. In the revision, we restructured the whole manuscript, added a new section to introduce the framework of exploring place-informative objects and scenes, conducted a follow-up experiment to discuss the new findings about geo-bias in image database, and added a new discussion section to highlight the findings of this work.

We paid more attention to the description of methodology, balancing the detail and abstraction to make the article available for readers with corresponding interest. Moreover, we add a new section in the revision to introduce the proposed general data-driven framework of exploring place-informative objects and scenes from images.

Appendix B

Associate Editor Comments to Author (Dr Cecilia Mascolo):

There are still a number of presentation issue related to how the work is positioned and how the additional evaluation is described which need fixing. I will check that the authors have fixed this before the paper can be accepted.

Reviewer comments to Author:

Reviewer: 1

Comments to the Author(s)

I do acknowledge the effort of the authors in improving the paper. However, there is little remedy to add novelty to a contribution which has very little novelty. I am fine in seeing this paper accepted because it is technically correct but only if the authors agree on explicitly acknowledging the limitations of their contribution. This would be done with the following two actions:

1) The workflow is indeed better explained now, which is good. Yet, a better explanation doesn't make it any more novel. The authors simply don't make a methodological contribution in this space and this has to be acknowledged. Sentences like "This work makes a contribution to learning the visual features of places by proposing a DCNN and geo-tagged images based framework" and "The goal of this study is to build a data-driven framework to learn and compare the visual appearance of different places by exploring place-informative scenes and objects" need to be either removed or rephrased. What the authors do is to ****use an existing visual processing framework to extract distinctive visual aspects of cities**** -- this has to be clearly stated.

Thanks for the reviewer's comments. We have rephrased the relevant presentation in the introduction section, namely:

1. From "This work makes a contribution to learning the visual features of places by proposing a DCNN and geo-tagged images based framework",

To "This work makes a contribution to learning the visual features of places with a DCNN and geo-tagged images based **method built upon previous studies.**"

2. From "The goal of this study is to build a data-driven framework to learn and compare the visual appearance of different places by exploring place-informative scenes and objects",

To "The goal of this study is to **extract** distinctive visual aspect of places by **exploring** place-informative scenes and objects."

2) I don't think it is sufficient to cite the work of Santani and Doersch. In the related work or in the introduction (or in any other relevant part of the manuscript) I would like to see one or two paragraphs discussing explicitly those works, acknowledging how this paper methodologically builds upon them and highlighting the differences.

Thanks for the reminder. We have added a detailed description in the introduction section on the works that are most relevant to this study. Details are shown below:

“Similar works have been done in recognizing geo-locations [26-28], detecting landmarks [11, 12, 29-31], examining ambiance perceptions [32, 33], and identifying urban identities [4, 5, 34].”

“The work closest to ours are Doersch (2012) and Zhou (2014). Doersch (2012) proposed a framework based on discriminative clustering to discover the clusters of local image patches that make a city distinct. These patch clusters reflect very local cues about the urban properties, such as windows style or building textures, while our work focus on higher-level concepts such as scenes and objects. Zhou (2014) only analyzed distinct scenes across cities using a scene classification framework, while our work unifies the analysis of scenes and objects.”

I am sorry but I did not understand the purpose of the additional experiment reported. I would encourage the authors to make an additional effort to motivate it better.

Thanks for pointing this out. The additional experiment in the discussion section is about the way to increase the accuracy of city recognition task. Since the experiment is not quite relevant to either “learning place-informative scenes and objects” or “measuring place distinctiveness and similarity”, which is the main topic of this paper, after our careful consideration, we decide to remove this irrelevant discussion.

Regarding the application on Panoramio photos vs. other sources: the application on Panoramio does not make the paper wrong but it limits its interestingness, in my opinion, which will be probably reflected in a reduced impact of the manuscript in the community. That said, I am fine with it as long as the potential limitations are mentioned.

Thanks for the reviewer’s suggestion. In the revision, we added a subsection to discuss the potential limitation of this study. Details are shown below:

“The performance of the proposed framework depends largely on the representativeness of the data source used. Crowd-sourced data is always biased towards a group of people that generates the data. For instance, the Panoramio dataset used in this study are mainly contributed by tourists and photographers, who will have a different perspective of a city's visual feature from the local residents. A single data source is therefore difficult describing the objective visual aspects of a city. Future studies are expected to integrate multiple data sources, e.g., Google Street View imagery, to represent the place comprehensively.”